# Uptake and Patient Perspectives on Additional Testing for Novel Disease-Associated Genes: Lessons from a PAH Cohort

**DOI:** 10.3390/genes12101540

**Published:** 2021-09-28

**Authors:** Samara M. A. Jansen, Lieke M. van de Heuvel, Arjan C. Houweling, J. Peter van Tintelen, Frances S. de Man, Anton Vonk Noordegraaf, Harm Jan Bogaard

**Affiliations:** 1Department of Pulmonary Medicine, Amsterdam University Medical Center (UMC) (Vrije Universiteit), 1081 HV Amsterdam, The Netherlands; s.jansen1@amsterdamumc.nl (S.M.A.J.); fs.deman@amsterdamumc.nl (F.S.d.M.); a.vonk@amsterdamumc.nl (A.V.N.); 2Department of Human Genetics, Amsterdam University Medical Center (UMC) (Vrije Universiteit), 1081 HV Amsterdam, The Netherlands; l.m.vandenheuvel@amsterdamumc.nl (L.M.v.d.H.); a.houweling@amsterdamumc.nl (A.C.H.); J.P.vanTintelen-3@umcutrecht.nl (J.P.v.T.); 3University Medical Centre Utrecht, Department of Genetics, Utrecht University, 3584 CX Utrecht, The Netherlands

**Keywords:** uptake testing, pulmonary arterial hypertension, re-contacting

## Abstract

Background: Pulmonary arterial hypertension (PAH) has an identifiable genetic cause in 5% of all PAH cases. Due to health benefits conferred by the early detection of PAH and the recent identification of additional PAH-associated genes, we decided to offer (extended) genetic testing to all incident and prevalent idiopathic PAH (iPAH) and pulmonary veno-occlusive disease (PVOD) patients in our clinic. Here, we report the lessons learned from (re-)contacting iPAH/PVOD patients concerning the uptake and analysis of identified PAH-associated genes and patient perspectives of the approach. Methods: Between January 2018 and April 2020, all iPAH/PVOD patients who were not previously genetically tested (contact group) and those who tested negative on prior analysis of *BMPR2* and *SMAD9* variants (re-contact group) were (re-)contacted for (additional) genetic testing. Results: With our approach, 58% of patients (84 out of 165) opted for genetic counselling, and a pathogenic variant was found in 12% of cases (*n* = 10) (re-contact group, 11%, and contact group, 13%). Eighty-six percent of participants of the survey study appreciated being (re-)contacted for genetic testing. Mild psychosocial impacts were observed. Conclusions: Our report shows the importance of (re-)contact and interest of patients (as indicated by the uptake, mild psychosocial impact and appreciation) in PAH.

## 1. Introduction

Pulmonary arterial hypertension (PAH) is a severe condition characterized by pulmonary vascular remodelling and ultimately right heart failure [1]. Approximately five percent of all PAH cases are reported to have an identifiable genetic cause and are classified as hereditary PAH (hPAH) [1,2]. hPAH generally follows an autosomal dominant inheritance pattern with incomplete penetrance [3]. When genetic causes and associated conditions, such as connective tissue disease, congenital heart disease or portal hypertension have been excluded, PAH is classified as idiopathic (iPAH) [1]. Pulmonary veno-occlusive disease (PVOD) is a pulmonary vasculopathy grouped together with PAH which also occurs in both hereditary and non-hereditary forms. PVOD has similar clinical and hemodynamic characteristics as iPAH/hPAH but a different pathophysiology and worse prognosis [4,5].

We recently identified a (likely) pathogenic variant in 25% of patients in a Dutch cohort of prevalent PAH patients without an associated condition [6]. The vast majority (81%) of these patients had a variant in bone morphogenetic protein receptor type 2 (*BMPR2*), which is known to be the main genetic cause of hPAH [7]. Variants in the *BMPR2* gene have also been reported to cause PVOD [8], although heritable PVOD is mainly caused by variants in *EIF2AK4* [9]. Here, we report the lessons learned from re-contacting iPAH/PVOD patients concerning the uptake and analysis of recently identified disease-associated genes and patient perspectives of the approach. Our aim was to improve the prognosis of hPAH by the identification of (likely) pathogenic variants in iPAH/PVOD patients and their relatives at risk. The identification of pathogenic variants allows for the early detection and timely treatment of hPAH in genotype-positive relatives [10,11].

## 2. (Re-)Contact Procedure

### 2.1. Methods

Until 2018, genetic counselling and testing at the Amsterdam UMC PH centre was not routinely offered to sporadic PAH patients and was restricted to the sequencing of *BMPR2* and *SMAD9* genes. Given the increasing evidence of health benefits conferred by the early detection of PAH and the recent identification of additional genes associated with hPAH/PVOD, in 2018, we decided to offer genetic counselling and (extended) genetic testing to all incident and prevalent IPAH/PVOD patients in our clinic. A “(re)contact” program was set up to inform all patients about the possibility of (additional) genetic testing in the period between January 2018 and April 2020. In this program, all iPAH/PVOD patients in our cohort who were not previously genetically tested and those who tested negative on prior analysis for *BMPR2* and *SMAD9* variants were (re-)contacted for (additional) genetic testing. Diagnosis and classification of iPAH and PVOD in all patients followed current guidelines [1]. In short, PAH is defined as an increase in mean pulmonary arterial pressure (mPAP) ≥ 25 mmHg, pulmonary vascular resistance ≥ 3 WU and a pulmonary arterial wedge pressure (PAWP) ≤ 15 mmHg at rest, assessed via right heart catheterization. iPAH is diagnosed when associated conditions, such as connective tissue disease, HIV, drug abuse, congenital heart disease or portal hypertension have been excluded. The diagnosis PVOD can be established with a high probability based on the combination of clinical suspicion, decreased pulmonary function tests (i.e., decreased diffusion capacity of the lung for carbon monoxide (DLCO), arterial blood gases (severe hypoxaemia) and high-resolution computed tomography of the chest (septal lines, centrilobular ground-glass opacities/nodules and mediastinal lymph node enlargement). In some cases, a lung biopsy is needed to confirm a histological diagnosis of PVOD [1,12].

iPAH/PVOD patients, all unrelated, were initially (re-)contacted via an information letter from their treating physician (Figure 1). This letter included information about the possible consequences of an abnormal test result for themselves and for their relatives, who could potentially be at risk of PAH and therefore eligible for screening. An example of the information letter to index patients is given in Appendix A. The letter includes information for patients who had not previously been tested (DNA-) and for those who tested negative on prior analysis for *BMPR2* and *SMAD9* variants (DNA+, MUTATION-). One week after receiving this information letter, a psychologist (L.M.v.d.H.) from the Department of Clinical Genetics or a medical doctor (S.M.A.J.) from the Department of Pulmonary Medicine called the patient to provide additional information and to answer questions about genetic testing. Interested iPAH/PVOD patients received genetic counselling and could opt for genetic testing. An NGS panel was analysed, including 19 PAH-associated genes (*ABCA3, ACVRL1, BMPR1B, BMPR2, CAV1, EIF2AK4, ENG, FOXF1, GDF2, KCNA5, KCNK3, NOTCH1, NOTCH3, RASA1, SMAD1, SMAD4, SMAD9, TBX4*, and *TOPBP1*) as previously described [6]. When a (likely) pathogenic variant was detected, the patient was informed by the clinical geneticist about the identified genetic cause and its implications. In addition, the patient was encouraged to inform first-degree relatives about the possibility of genetic counselling and predictive DNA-testing. Patients with a (likely) pathogenic variant received a personalized letter to support them in informing their relatives about hPAH/PVOD, their potential risk, and how to access genetic counselling. hPAH patients in whom a pathogenic *BMPR2* variant was detected prior to January 2018 were also informed about current screenings options for their at-risk relatives via a family information letter. An example of the family information letter is given in Appendix A. Relatives could opt for genetic counselling and genetic testing when interested.

### 2.2. Results

The results of genetic testing were recently published [6]. Out of 165 iPAH/PVOD patients (119 contact and 46 re-contact) who were eligible for inclusion at our centre, 146 patients were (re-)contacted (Figure 2). A total of 84 patients (58%, 56 contact and 28 re-contact) pursued (additional) genetic counselling and testing. The median age at counselling was 60 years old (range 18–78 years), 61 out of these 84 patients (73%) were female, 52 patients (62%) who pursued counselling had an NYHA functional class 3 or 4 and 29 patients (35%) had a family history of cardiovascular diseases or lung diseases. Additional genetic testing revealed (likely) pathogenic variants in 10 patients (12%): re-contact group *TBX4* (*n* = 3) and contact group *BMPR2* (*n* = 4), *GDF2* (*n* = 2) and *EIF2AK4* (*n* = 1). In 11 out of 84 patients (9%), genetic testing revealed a variant of unknown significance (VUS) class 3: *NOTCH3* (*n* = 4), *BMPR2* (*n* = 3), *FOXF1* (*n* = 3) and *TBX4* (*n* = 2). One patient had two VUSs (*FOXF1* and *BMPR2*). After the initial identification of pathogenic variants in 10 iPAH/PVOD patients, 10 out of 42 relatives of these patients were shown to be carriers. These relatives carried a (likely) pathogenic variant in *TBX4* (*n* = 4), *BMPR2* (*n* = 4) and *GDF2* (*n* = 2).

## 3. Patient Perspectives

### 3.1. Methods

We administered two surveys to explore how patients responded to our (re-)contact approach, to evaluate its psychosocial impact and to assess whether our approach could be improved. The study was approved by the Medical Ethical Review Committee of the Amsterdam UMC, Vrije Universiteit (approval number 2017.541), and was conducted in accordance with the principles of the Declaration of Helsinki. Informed consent was obtained from all subjects. The surveys were sent out immediately after the telephone contact by the psychologist or medical doctor (T1) and after genetic counselling (T2, four months after T1). The surveys addressed the sociodemographic characteristics of subjects (i.e., age, education, ethnicity, and household composition), their health status (i.e., health complaints due to PAH and PAH treatment), satisfaction with the offer for additional DNA-testing (before and after genetic counselling), the psychosocial impact of our approach (i.e., anxiety and worries for themselves and/or their relatives) and their ideas on whether and how to improve our (re-)contact approach. After genetic counselling (T2), patients were asked whether their opinion on the (re-)contact approach had changed. To evaluate the psychological impact, adapted versions of validated questionnaires (the Cancer Worry Scale (CWS) and the Hospital Anxiety Depression Scale (HADS)) were administered [13,14].

### 3.2. Results

Initially, 50 out of 165 iPAH/PVOD patients provided informed consent to participate in the survey study. In total, 35 iPAH patients (70%) returned the survey at T1 and 28 iPAH patients (56%) returned the survey at T2. At T1, 18 out of 35 patients (51%) had previously been tested negative for variants in the *BMPR2* and *SMAD9* genes. Seventy-one percent were females with a median age of 64 years (range 29–76 years). Six patients did not pursue genetic counselling. Of these six patients, three were female, the median age was 65 years (range 55–73 years), two patients had previously been tested negative, five of them had children and three patients had no/few complaints. Twenty-nine out of the thirty-five patients that returned the survey (83%, seventeen contact and eighteen re-contact) opted for genetic testing after (re-)contact and only one patient tested positive for a pathogenic variant. Most respondents had children (*n* = 30, 86%). Regardless of the results of genetic testing, almost all participants appreciated being (re-)contacted for (additional) genetic testing (*n* = 30, 86%): see Figure 3. Almost all patients reported being satisfied about the way they were informed and felt free to decide whether they would undergo (additional) genetic testing or not (94% and 97%, respectively). We observed fairly high anxiety scores in patients after re-contact (T1, HADS anxiety mean score 7.9 ± 2.2). Depression and worry scores were within normal ranges (HADS depression mean score of 6 ± 1.1 and CWS mean score of 11.1 ± 2.9). After disclosure of the genetic test results (T2), we observed similar levels (HADS anxiety mean score: 8.4 ± 2.0, HADS depression mean score: 6.1 ± 1.6 CWS mean score: 11.3 ± 3.2). A minority of patients (23%) thought that the (re-)contact approach could be improved (Figure 3). Seventeen percent of patients would rather have had more information about the advantages and disadvantages of genetic testing during genetic counselling or via the information letter before deciding to undergo (additional) genetic testing. Improvements that were suggested were information provision via phone/video call by a specialist, such as a clinical geneticist/pulmonologist (*n* = 2) or specialised PAH nurse (*n* = 1) instead of a psychologist or general doctor, and information provision at the outpatient clinic instead of a letter and phone call (*n* = 1).

## 4. Discussion

Although the advantages and disadvantages of re-contact genetic approaches have been described previously [15,16], there are not yet any recommendations or guidelines for re-contacting patients for additional genetic testing after the identification of novel disease-associated genes. This report gives a suggestion on how to set up a (re-)contact procedure. (Re-)contact can reveal more hereditary PAH/PVOD cases who were originally diagnosed as sporadic iPAH/PVOD. With our approach, 58% of patients opted for genetic counselling and a pathogenic variant was found in 12% of cases (re-contact group 11% and contact group 13%). However, more research is needed to optimize this (re-)contact procedure.

Identification of a genetic cause in PAH patients is important for the early detection of PAH in at-risk relatives. Our (re-)contact approach identified 10 relatives at risk as carriers and these carriers will be screened annually to detect PAH in early stages. Asymptomatic *BMPR2* mutation carriers have a significant risk of 2.3% per year of developing PAH (0.99% per year in males and 3.5% per year in females) [11]. As shown in the DELPHI-2 study of Montani et al., annual screening of unaffected mutation carriers could be beneficial: annual screening identified five PAH cases and all patients remained at low risk with oral PAH-specific therapy during follow-up [11]. These findings highlight the importance of detecting pathogenic variants in PAH patients and their relatives at risk. However, more research and multi-centre collaborations are needed to gain better insights in hereditary PAH, searching for the “second” hit and to improve early detection of PAH.

## 5. Conclusions

As expected, (re-)contact revealed new hereditary PAH/PVOD cases and genotype-positive relatives. Novel disease-associated genes are identified at an unprecedented rate. Therefore, additional genetic testing and re-contacting patients is not only relevant for PAH, but also for patients with other (potential) genetic diseases who have undergone limited testing in the past, especially when early detection and treatment options provide health benefits. Our report shows the importance of (re-)contact and interest of patients (as indicated by the uptake, mild psychosocial impact and appreciation) in PAH. Requirements for additional re-contacting in the future are a registration of screened genes in patients analysed for a potential genetic disease and generic informed consent to re-contact patients if needed.

## Figures and Tables

**Figure 1 genes-12-01540-f001:**
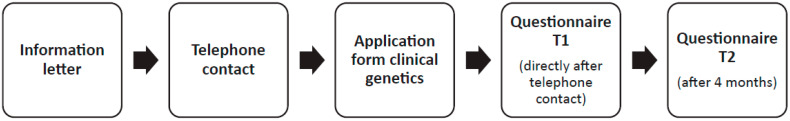
A schematic overview of the (re-)contact approach in idiopathic pulmonary arterial hypertension (iPAH) patients and pulmonary veno-occlusive (PVOD) patients. All iPAH/PVOD patients in our cohort who were not previously genetically tested (CONTACT group) or who tested negative on prior analysis of the *BMPR2* and *SMAD9* genes (Re-CONTACT group) were (re-)contacted for (additional) genetic testing.

**Figure 2 genes-12-01540-f002:**
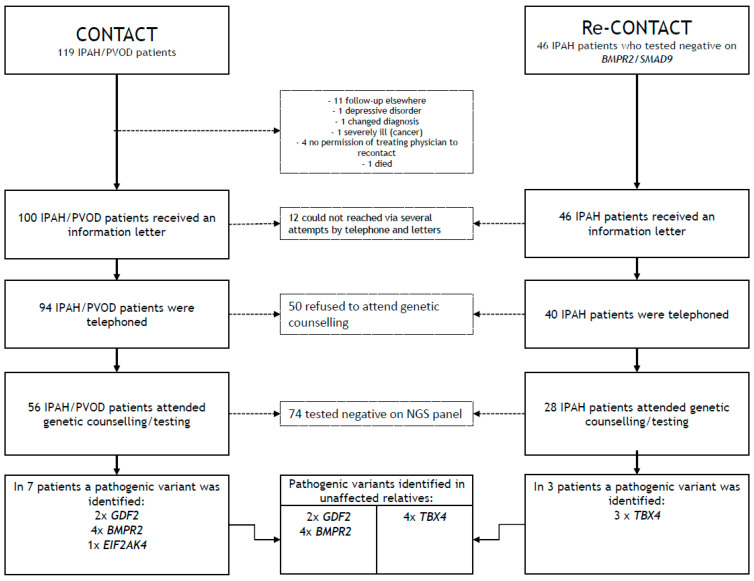
Flowchart of the (re-)contact approach with patient inclusion and results of genetic testing. The (re-)contact approach consisted of iPAH/PVOD patients who were not previously genetically tested (CONTACT group) or who tested negative on prior analysis of the *BMPR2* and *SMAD9* genes (Re-CONTACT group). iPAH: idiopathic pulmonary arterial hypertension, PVOD: pulmonary veno-occlusive disease, NGS: next-generation sequencing.

**Figure 3 genes-12-01540-f003:**
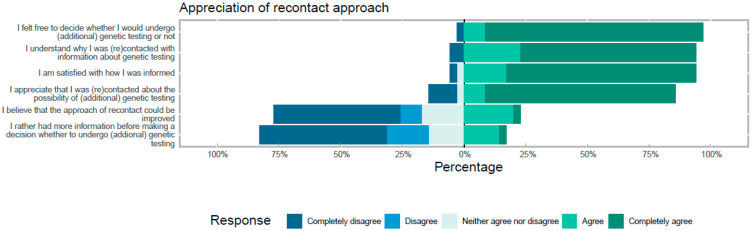
Patients’ experiences of the re-contact approach. All responses are shown in percentages and divided in subcategories.

## Data Availability

Not applicable.

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
