# Peer review of "Uptake and Patient Perspectives on Additional Testing for Novel Disease-Associated Genes: Lessons from a PAH Cohort"

_genes, 2021, doi:10.3390/genes12101540_

Round 1

Reviewer 1 Report

Well written commentary.

Minor comments/proposals only:

In general it is very good idea to address relatives at risk by letter spread by "index" patients (definitely beyond PAH) - if the letter could be published (abbreviated) could be helpful for readers/future users of this method. Respondence rate (58 % ?) could be presented more clearly/more highlighted together with its determinants (in addition to children, ... symptoms of disease, education, ...); the information is there, but it is not easy to extract it easily. In addition, description of characteristics of non-compliant/non-adherent patients/individuals (if any)could add valuable information.

Author Response

Dear reviewer,

Thank you very much for the careful assessment of our short communication entitled " Uptake and patient perspectives on additional testing for novel disease-associated genes; lessons from a PAH cohort”. We very much appreciate the constructive criticisms from the reviewers of Genes and believe that by incorporating the suggestions as given by the reviewers, the quality of our communication has improved significantly. Please find the point-by-point response to all of the issues raised by the reviewers. We are looking forward to receiving your reaction to this revised manuscript.

Comments and Suggestions for Authors

Well written commentary.

Minor comments/proposals only:

C1. In general it is very good idea to address relatives at risk by letter spread by "index" patients (definitely beyond PAH) - if the letter could be published (abbreviated) could be helpful for readers/future users of this method.

R1. We thank the reviewer for this suggestion. We agree that adding this letter to address relatives could be helpful as an example for future research or (re-)contact procedures. We have added this letter as well as the letter to the index patients for the (re-)contact procedure as supplemental material to our communication.

C2. Respondence rate (58 % ?) could be presented more clearly/more highlighted together with its determinants (in addition to children, ... symptoms of disease, education, ...); the information is there, but it is not easy to extract it easily.

R2. We agree with the recommendation of the reviewer. The reason for the unclearness was that all determinants presented in the short communication were part of the survey study which is a sub-study of our re-contact procedure. To further clarify this, we have added these determinants of the patients who pursued counselling; page 3, line 94-97.

C3. In addition, description of characteristics of non-compliant/non-adherent patients/individuals (if any)could add valuable information

R3. We agree with the reviewer that characteristics of the non-adherent patients could be valuable for the improvement of the re-contact procedure. However, we do not have these characteristics of all 50 patients who refused to attend genetic counselling (figure 2), since they did not gave informed consent. We added the available patient characteristics information of non-attenders to the survey study. (page 4, line 130-133).

Reviewer 2 Report

Overall, this Communication article is in a concept to describe the uptake and patient perspectives on additional testing for novel disease-associated genes in the PAH cohort. The work is an addition to the PVOD/PAH field but needs more information to add.
A number of major concerns are outlined below. 
1. Please add more wording on how did your cohort confirm PVOD/PAH for these cases you used in this study. As we know, PVOD is difficult to diagnose. 
2. In the Flowchart of Figure 2, 7 out of 119 PVOD/PAH patients have a pathogenic variant was identified in the final: 2XGDF2, 4XBMPR2, 1XEIF2AK4; 3 IPAH patients have a pathogenic variant was identified: 3XTBX4. These two groups have totally different identified pathogenic variants. It is uncertain why are you pooling two groups together In the block of 10 unaffected mutation carriers.

Author Response

Dear reviewer,

Thank you very much for the careful assessment of our short communication entitled " Uptake and patient perspectives on additional testing for novel disease-associated genes; lessons from a PAH cohort”. We very much appreciate the constructive criticisms from the reviewers of Genes and believe that by incorporating the suggestions as given by the reviewers, the quality of our communication has improved significantly. Please find the point-by-point response to all of the issues raised by the reviewers. We are looking forward to receiving your reaction to this revised manuscript.

Comments and Suggestions for Authors

Overall, this Communication article is in a concept to describe the uptake and patient perspectives on additional testing for novel disease-associated genes in the PAH cohort. The work is an addition to the PVOD/PAH field but needs more information to add.

A number of major concerns are outlined below. 
C1. Please add more wording on how did your cohort confirm PVOD/PAH for these cases you used in this study. As we know, PVOD is difficult to diagnose. 

R1. We thank the reviewer for this suggestion. We agree that PVOD is difficult to diagnose. We have added a section in the short communication on the diagnosis and classification of iPAH and PVOD in our cohort (page 2, line 49-60).

C2. In the Flowchart of Figure 2, 7 out of 119 PVOD/PAH patients have a pathogenic variant was identified in the final: 2XGDF2, 4XBMPR2, 1XEIF2AK4; 3 IPAH patients have a pathogenic variant was identified: 3XTBX4. These two groups have totally different identified pathogenic variants. It is uncertain why are you pooling two groups together In the block of 10 unaffected mutation carriers.

R2. We agree with the reviewer that pooling two groups of unaffected mutation carriers together in this figure makes it unclear. We have now separated these two groups in this figure to clarify this.

Round 2

Reviewer 2 Report

Please check the description of figure 2 on page 3 of 6, lines 102-105.

Please check the patient number of IPAH BLOCK: in 3 patients a pathogenic variant was identified 3x TBX4, but in the final block: unaffected mutation carriers were identified: 4x TBX4. it is not consistent in the number of the patient.